# Assessing breast cancer awareness on reproductive age women in West Badewacho Woreda, Hadiyya Zone, South Ethiopia; Community based cross- sectional study

**Mengistu Lodebo Funga◯\*, Zerihun Damissie Dilebo, Anebo Getachewu Shuramo, Tessema Bereku**

Midwifery Department, Hosanna College of Health Sciences, Hossana, Ethiopia

\* mengestulodebo@gmail.com

## Abstract

### Introduction

Breast cancer is the most commonly diagnosed cancer in women worldwide, in both high- and low-income countries. Individual and community awareness of breast cancer can be extremely beneficial. However, breast cancer awareness is extremely low among Ethiopian women, particularly in rural areas. Thus, the aim of this study was assessing awareness of breast cancer on reproductive-aged women in West Badewacho Woreda, Hadiya Zone, South Ethiopia, 2020.

### Method

A community-based-cross sectional study was employed from April 18 to May 16, 2020. To obtain data, a pre-tested structured interviewer-administered questionnaire was employed. A multistage random sampling technique was employed to select reproductive-aged women from Woreda. Data was entered into a computer using Epi data version 3.1, and edited, cleaned and analyzed using SPSS windows version 20. Bivariable and multivariable analyses were used to identify determinants related to awareness of breast cancer.

### Result

A total of 578 reproductive-age women participated in this study. Only 364 women of reproductive-age (63%) had heard about breast cancer. 234 (64.3%) of the respondents were aware of breast cancer. After adjusting for other variables, husbands' educational status [AOR = 0.262; 95% CI (0.076, 0.900)], family history of breast cancer [AOR = 0.281; 95% CI (0.132, 0.594)] and having TV/Radio [AOR = 0.489; 95% CI (0.246, 0.972)] were significant predictors for awareness of breast cancer.

**Data Availability Statement:** All relevant data are in the paper and its supporting information.

**Funding:** MLF received fund from Hossana College of Health Sciences for the research project under the grant number 6223. We acknowledge the college for its close follow up and deciding to publish/disseminate the research. The college had no role in study design, data collection and analysis, or preparation of the manuscript.

**Competing interests:** The authors have declared that no competing interests exist.

**Abbreviations: CI**, Confidence interval; **SPSS**, Statistical package for social sciences; **BSE**, Breast Self Examination; **CBE**, Clinical Breast Examination.

## Conclusion

This study emphasized the importance of raising breast cancer awareness among women in the study area.

## Introduction

Breast cancer is one type of cancer. It is a malignant tumor that develops from breast tissue, which contains milk-producing glands called lobules and ducts that connect the lobules to the nipple [1, 2].

The specific cause for breast cancer is unknown, but it is considered a disease mainly associated with some risk factors. Being a female, age, family history of breast cancer, especially a first-degree relative, early menarche at the age of 12 and under, late menopause after the age of 55 years, and having the first child after the age of 30, prolonged use of oral contraceptive pills, hormonal therapy, and being overweight or obese and null parity are risk factors. Breast milk, on the other hand, appears to be protective against breast cancer [3–5].

Breast cancer is the most commonly diagnosed cancer in women (24.2%, or roughly one in every four new cancer cases diagnosed in women worldwide), affecting 2.1 million women each year and causing the greatest number of cancer-related deaths in women. In 2019, an estimated 268,600 new cases of invasive breast cancer will be diagnosed among women and 41,760 women are expected to die from breast cancer. While breast cancer rates are higher among women in more developed regions, rates are increasing in nearly every region globally [6, 7].

Breast cancer is a disease that affects not only affluent countries, but also low- and middle-income countries. In reality, nearly sixty percent of women who die from breast cancer live in developing nations, where survival rates might be as low as 20%. Similarly, breast cancer is the most common malignancy among Ethiopian women. Many occurrences go unreported because women in rural regions sometimes seek treatment from traditional healers before seeking medical help [8]. Women in underdeveloped nations are more likely to be diagnosed late in breast cancer and in many situations due to weak health care systems and limited access to early detection and treatment; even access to supportive and palliative care is limited [9].

Though, the Ethiopian government invests a lot in the health sector, communicable, non-communicable and infectious diseases are still the major public health issues in the country [10].

The poor awareness, wrong beliefs and present for care at a late stage in the disease of breast cancer among women, where made treatment is most ineffective, negative perception of the curability of a cancer detected early [3, 11].

Promotion of women's awareness about breast cancer and its symptoms can encourage them to perform better and timelier breast self-exams, but lower awareness about breast cancer in rural women reduces their chance of detecting breast cancer symptoms, causing the breast cancer to be detected at an advanced stage [12]. Because the majority of Ethiopian women live in rural areas, they face delays in receiving care and, as a result, are more likely to be diagnosed with late stages of breast cancer. As a result, there is an immediate need to raise breast cancer awareness levels. Despite the aforementioned facts, little was known regarding breast cancer awareness among reproductive-aged women in the southern part of Ethiopia. Therefore, the aim of this study was assessing awareness of breast cancer on reproductive-aged women in Hadiya Zone, West Badewacho Woreda, 2020.

## Materials and methods

### Study design, setting and sampling

A community-based cross-sectional study design was employed from April 18 to May 16, 2020 in West Badewacho woreda in Hadiyya Zone, Southern Nations, Nationalities and People's Region, Ethiopia. The Woreda is situated 357 km southwest of Addis Ababa and 127 km from the regional capital, Hawassa. The Woreda has 20 rural and 2 urban kebeles that are arranged into 180 gotti (mahandar). As for the health infrastructure situation in Woreda, there are four health centers and 22 health posts that provide basic health care. In 2020, the woreda's population was predicted to be 113,265 people. The overall number of households in the woreda was estimated to be 23,115, with 26,391 women of reproductive age (23.3%).

The source population was all reproductive-aged women. The study population consisted of reproductive-aged women who were randomly selected at a household level from the source population. Women who were not permanent inhabitants of the Woreda (less than six months), had significant sickness during data collection, or were under the age of 18 were excluded from the study.

The sample size of this study was calculated using a single population proportion formula, taking proportion of awareness/ knowledge of breast cancer in the Bale Zone [11], margin of error, confidence level, design effect and non-response rate were assumed to be 56.2%, 5%, 95%, 1.5 and 5%, respectively.

No previous study, to the best of our knowledge, has determined the proportion of the population in the study area who is aware of breast cancer. We used research from other areas, and because the variation in the study population was expected to be high, we needed a large sample size to detect a difference. In contrast, we used at least two stages lower in the sampling process to arrive at the final sampling unit. As a result, we used a design effect of 1.5 to multiply our sample size in order to minimize variability and detect the effect observed regarding breast cancer awareness.

$$n = \frac{(Z\alpha/2)^2 p(1-p)}{d^2}$$

Where: ni = Sample size; Z $(\alpha/2)^2$ = confidence level; p = proportion of awareness (0.562); d = marginal of error

$$n = \frac{(1.96)^2 (0.562)(1-0.562)}{(0.05)^2}$$

n = 378 Individuals Using design effect 1.5 i.e. 378*1.5 = 567 Thus, finally taking 5% non-response rate the final sample size was **595**.

A multistage random sampling technique was employed. Initially, stratification of woreda into urban and rural was done. Then all the rural and urban Kebeles in the woreda were listed separately in the frame. Then, seven rural and one urban kebele were selected randomly using a lottery method. Proportional allocation was done to each selected kebele depending on the size of the houses containing the eligible population. A list of gotts (mahandar) was made for each kebele from the randomly picked kebele. The gotts (mahandar) were then selected using lottery techniques. As a sampling frame, the health extension works were used to extract a list of households containing eligible reproductive-aged women in selected kebele at each gott (mahandar) level. Again, a proportional allocation was made based on the number of eligible households in each selected gott from each selected kebele. The households with eligible reproductive-aged women were then selected from the sampling frame using a simple random

method. Then, with the assistance of health extension workers, data collectors went to each household, using the name of the head of the household as a guide. Finally, the lottery method was used to select one woman interviewee whenever there were two or more women of child-bearing age (18–49 years) in the selected household.

## Data collection method

A pretested structured questionnaire was used to collect data from each study participant. The questionnaire was adapted from related literatures [9, 11, 13] with minor changes to fit the objectives of this study and the local context.

Seven diploma nurses were recruited as data collectors, with three BSc nurses recruited as supervisors. The overall data collection procedure plan would be followed by the principal investigator.

The Ahmaric version of a structured questionnaire was used in a face-to-face interviewer administered method of data collection.

## Study variables

**Dependent variable.**   Breast cancer awareness
**Independent variables.**   *Socio-demographic characteristics*

➢ Age

➢ Residences

➢ Marital status

➢ Woman educational status

➢ Woman occupation

➢ Income

➢ Religion

➢ Ethnicity

➢ Husband educational level

   *Communication related variable.*

➢ Having TV/Radio

   *Past history characteristics.*

➢ Family history of breast cancer

## Operational definition and terms

- **Awareness of breast cancer**- was measured by asking questions about breast cancer awareness (risk factor, sign &symptoms, preventive measures, screening methods and treatment). Each correct answer was scored 'Yes' and each incorrect answer was scored 'No'. The mean score was used to calculate the cumulative mean score of participants' breast cancer awareness. Those who scored higher or equal to the mean value were considered "**aware**," while those who scored lower than the mean were considered "**unaware**" of breast cancer. We used mean to categorize the awareness after reviewing previous research [13, 14].

## Data quality assurance

Two-day training was provided to the data collectors and supervisors. The questionnaire was first prepared in English, translated into Ahmaric and then it was re-translated back to English to check for its consistency. A week before the actual data collection, investigators, supervisors, and data collectors pre-tested the Ahmaric version of the questionnaire in East Badewacho Woreda Ajeba and Wera Lalo kebeles, which have similar demographics to the research population. The appropriate changes were then made to standardize and ensure the validity of the document. The surveillance was done on a daily basis. Furthermore, the data was extensively cleansed before being carefully fed into the computer to begin the analysis.

## Data processing and analysis

The data was coded and entered into Epi-data version 3.1 before being exported to the statistical software SPSS version 20 for analysis. Variable recoding and computation were done as needed. To summarize the data, descriptive statistics such as frequency, percentage, mean, standard deviation, and range were used. A binary logistic regression analysis was carried out. All variables in the bivariable analysis with a P-value < 0.25 were candidates for multivariable logistic regression. The back-ward likelihood ratios of logistic regression were performed to identify the factors associated with breast cancer awareness. The model's goodness of fit was checked by the Hosmer-Lemeshow test and the p-value was found to be 0.723 (> 0.05), which revealed that the model was good. Adjusted odds ratios (AOR) at 95% CI were computed to measure the strength of the association between the outcome and the explanatory variables. P-value < at 0.05 was considered as statistically significant in the study in the multivariable model.

## Ethical consideration

Ethical clearance and permission was obtained from the Institutional Review Board (IRB) *(code 502/2012 E.C)* of Hosanna College of Health Sciences with protocol reference number by *HCHS 09/4452* and offered to Woreda Health Office, and then permission was obtained from study Kebeles. After a detailed explanation of the study's goal, expectations for participation, and potential risks and benefits to each participant, informed oral agreement was obtained. Voluntary participation and responses were kept confidential and anonymous. Women were interviewed at their home in a private area and no family members were allowed to accompany the participants to permit freedom of expression, privacy, and confidentiality. According to the ethical criteria of the college's institutional review board, we documented participant permission with the recorded data for at least two years.

# Results

The interview was completed by 578 reproductive-aged women out of a total of 595 reproductive-aged women, yielding a response rate of 97%. The analysis did not include the seventeen incomplete questionnaires. The results are presented under subheadings as follows:

## Socio-demographic characteristics of respondents

In this study, the age of participants ranged from 18 to 48 years. The mean age of respondents was 27.59±7.18 SD years. One hundred and eighty-two (31.5%) of study participants belonged to the age below 23 years. Four hundred and sixty-six (80.6%) of participants were rural. Regarding work status, 169 (29.2%) were farmers. One hundred and seventy-nine (31%) of the women were unable to read and write. Regarding religion, 434 (75.4%) of respondents were

Protestant religious followers. Five hundred and forty-two (93.8%) of the respondents were Hadiya by ethnicity. Regarding marital status, nearly two-thirds of the women (368, or 63.7%) were married. One hundred and fifteen of their husbands (31.3%) were unable to read and write. Concerning monthly income, the majority 434 (75.4%) of respondents were < 1311 ETB. Only 103 (17.8%) of respondents reported that they had a family history of breast cancer. Around 37% of study participants had access to television or radio [Table 1].

### General awareness of breast cancer

Three hundred and sixty-four reproductive-aged women (63%) of the respondents had heard of breast cancer. According to the operational definition, 130 (35.7%) of the respondents were unaware, i.e., scored less than the mean at (4.39 ±SD 4.27). But, 234 (64.3%) of respondents were aware, i.e., scored greater than or equal to the mean.

Breast lumps were mentioned by the majority of women (60.4%) as signs and symptoms of breast cancer. Family history was mentioned by nearly half of the respondents (51.1%) as a common risk factor for breast cancer. The majority (64.6%) of participants stated that breast cancer was preventable, and they reported regular screening (53.6%) as a common breast cancer preventive method. More than three-quarters (77.5%) of those who heard about breast cancer mentioned clinical breast examination as a common screening method for breast cancer, and 59.1% believed that breast cancer was treatable. More than half of the participants (68.4%) mentioned hormonal therapy as a major treatment method [Table 2].

### Sources of information

One hundred eighty-seven (51.4%) of those who had heard about breast cancer had heard from health workers, neighbors 138 (37.9%), and friends 133 (36.5%). In addition, 124 (34.1%) came from radio/TV, 75 (20.6%) from class, 55 (15.1%) from books/nets, and 16 (4.4%) from other sources such as the internet and magazines [Fig 1].

### Factors associated with awareness of breast cancer

During bivariable analysis, all variables with a p-value of less than 0.25 were entered into multi-variable logistic regression analysis. In the study, variables with p-values less than 0.05 in multivariate analysis were considered statistically significant. The variables that significantly predicted breast cancer awareness included the husband's educational status, a family history of breast cancer, and having TV/Radio.

In a multivariable logistic regression analysis, those whose husbands attended primary school were 74% less likely to be aware of breast cancer compared to those whose husbands attended tertiary school [AOR = 0.26; 95% CI (0.07, 0.90)]. Similarly, those with no family history of breast cancer were 72% less likely to be aware of the disease than those with a family history [AOR = 0.28; 95% CI (0.13, 0.59)]. In the same way, those who did not have access to television or radio were 51% less likely to be aware of breast cancer than their counterparts [AOR = 0.49; 95% CI (0.24, 0.97)] [Table 3].

### Discussion

Improved societal awareness of non-communicable diseases, particularly breast cancer and its predictors, is critical for the success of prevention interventions. As a result, the focus of this study was assessing breast cancer awareness on reproductive-aged women.

According to our study finding, 64.3% of study participants were aware of breast cancer based on the composite score for signs and symptoms, risk factors, prevention methods,

**Table 1. Socio-demographic characteristic of respondents in West Badewacho Woreda, Hadiya Zone, South Ethiopia, 2020 (n = 578).**

| Variables | Categories | Frequency | Percentage |
|---|---|---|---|
| Age | = <23 | 182 | 31.5 |
|  | 24–29 | 178 | 30.8 |
|  | 30–34 | 101 | 17.5 |
|  | = >35 | 117 | 20.2 |
| Residence | Rural | 466 | 80.6 |
|  | Urban | 112 | 19.4 |
| Work status | Farmer | 169 | 29.2 |
|  | Housewife | 95 | 16.4 |
|  | Student | 122 | 21.1 |
|  | Merchant | 85 | 14.7 |
|  | Daily laborer | 33 | 5.7 |
|  | Gov't employee | 74 | 12.8 |
| Educational level of women | cannot read and write | 179 | 31.0 |
|  | read and write | 131 | 22.7 |
|  | Primary | 109 | 18.9 |
|  | Secondary | 84 | 14.5 |
|  | higher or tertiary | 75 | 13.0 |
| Religion | Orthodox | 60 | 10.4 |
|  | Catholic | 67 | 11.6 |
|  | Muslim | 3 | .5 |
|  | Protestant | 434 | 75.1 |
|  | Others* | 14 | 2.4 |
| Ethnicity | Hadiya | 542 | 93.8 |
|  | Kambata | 18 | 3.1 |
|  | Wolayita | 11 | 1.9 |
|  | Others** | 7 | 1.2 |
| Marital status | Married | 368 | 63.7 |
|  | Single | 188 | 32.5 |
|  | Divorced | 12 | 2.1 |
|  | Widowed | 10 | 1.7 |
| Educational level of husband | cannot read and write | 115 | 31.3 |
|  | read and write(non-formal) | 98 | 26.6 |
|  | Primary | 57 | 15.5 |
|  | Secondary | 36 | 9.8 |
|  | higher or tertiary | 62 | 16.8 |
| Income | 1311 | 434 | 75.1 |
|  | >1311 | 144 | 24.9 |
| Family history of BC | No | 474 | 82 |
|  | Yes | 104 | 18 |
| Having TV/Radio | No | 362 | 62.6 |
|  | Yes | 216 | 37.4 |

**Others**: only Jesus*, Adventist*, Ahmara**, Silte**, Gurage**

common diagnosis methods, and treatment of breast cancer. This study agreed with studies conducted in Iran [12], central India [14], Nigeria [3], and Ethiopia [15]. This result is higher than that of a cross-sectional study conducted in Eastern China (18.6%) [16] and Saudi Arabia

**Table 2. General awareness about breast cancer in West Badewacho Woreda, Hadiya Zone, South Ethiopia, 2020.**

| Variables | Categories | Frequency | Percentage |
|---|---|---|---|
| Sign and symptoms* | Breast lump | 220 | 60.4 |
| | Breast pain | 202 | 55.5 |
| | Discharge | 139 | 38.2 |
| | Nipple retraction | 68 | 18.7 |
| | Redness and engorgement | 72 | 19.8 |
| | Itching | 110 | 30.1 |
| | Change in size of the breast | 37 | 10.2 |
| Risk factors* | Family history | 186 | 51.1 |
| | Contraceptive pills | 93 | 25.5 |
| | Increasing age/aging | 38 | 10.4 |
| | Being a woman | 162 | 44.5 |
| | Obesity | 23 | 6.3 |
| | Not breast feeding | 71 | 19.5 |
| | Early-onset of menarche | 48 | 13.2 |
| | Late menopause | 43 | 11.8 |
| | Smoking | 44 | 12.1 |
| | Alcohol | 49 | 13.5 |
| Breast cancer preventable (n = 364) | No | 129 | 35.4 |
| | Yes | 235 | 64.6 |
| By which method can it be prevented*? | Initiate breast feeding | 118 | 50.2 |
| | No smoking | 61 | 26 |
| | Not drinking alcohol | 58 | 24.7 |
| | Regular screening | 126 | 53.6 |
| | Physical Exercise | 13 | 5.5 |
| | Combat obesity | 9 | 3.8 |
| | Avoid OCP | 34 | 14.5 |
| | Wearing bra | 34 | 14.5 |
| What common screening methods are of breast cancer do you know?* (n = 364) | Breast self-examination | 168 | 45.3 |
| | Clinical breast examination | 283 | 77.5 |
| | Mammography | 40 | 11 |
| Is breast cancer is treatable? (n = 364) | No | 149 | 40.9 |
| | Yes | 215 | 59.1 |
| What is the treatment of breast cancer?* | Chemotherapy and radiotherapy | 57 | 26.5 |
| | Hormonal therapy | 147 | 68.4 |
| | Surgery or removal of the whole breast | 135 | 62.8 |

*more than one option is reported by a participant.

(39.7%) [17]. This discrepancy could be attributable to sample size, respondent inclusion criteria, data collection methods, total number of questions asked of respondents, and the construction and computation of the awareness item.

This study's findings are also lower than those of a previous study conducted in Benin (92.6%) [18]. This is most likely due to sample size, study setting, socio-cultural characteristics of respondents and the item's cut point score for awareness. In addition, participants received health education during multiple visits to the health facility to immunize their child.

Those whose husbands attended primary school were less likely to be aware of breast cancer than those whose husbands attended tertiary school. This finding was supported by research findings from Turkey [19] and China [20]. This is most likely due to the fact that people with

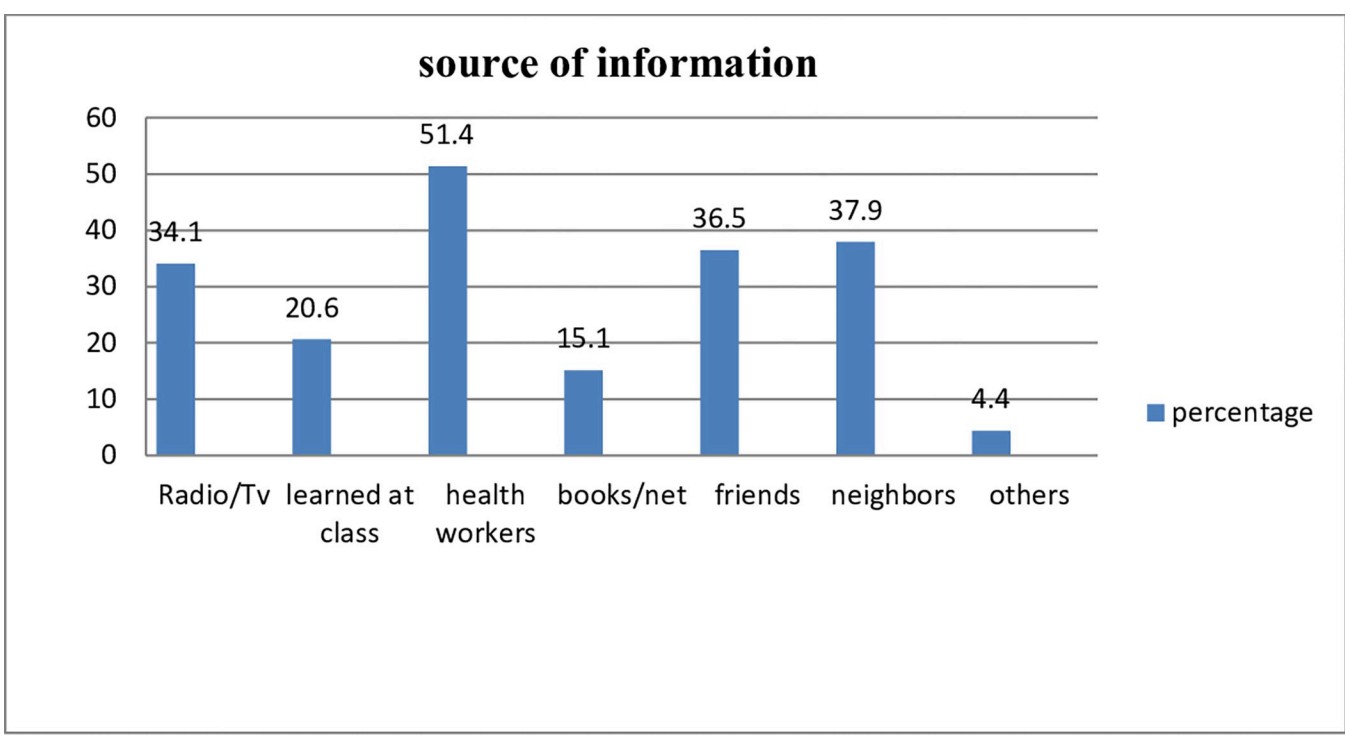

**Fig 1. Source of information for the awareness of breast cancer among reproductive-age women in West Badewacho Woreda, Hadiya Zone, South Ethiopia, 2020.**

higher levels of education are more capable of obtaining more and effective information from various sources, as well as having the opportunity to communicate with their families about these specific health problems. They also have a greater ability to equip their homes with various information sources and a higher income to do so.

This study revealed that a family history of breast cancer is associated with breast cancer awareness. Those who did not have a family history of breast cancer were less likely to be aware than their counterparts. This finding was consistent with previous research from Iran [21], China [16], and Ethiopia [15]. This could be because having a family history of breast cancer provided them with more information and increased their health-seeking behavior.

The current study showed a significant link between having TV/Radio and being aware of breast cancer. This finding was supported by research from Nigeria [22, 23] and Uganda [24]. It is possible that the mass media will play an important role in the development of awareness. The media are generally regarded as channels of communication capable of simultaneously reaching diverse audiences with the same messages. They also persuade and stimulate social mobilization. In other words, because of their ability to reach every segment of the community, the mass media can be regarded as powerful sources of information. They are able to spread messages about issues, ideals, and products. Furthermore, the media has the ability to raise awareness and knowledge about topics of interest. As a result, many scholars agree that awareness leads to knowledge, and knowledge leads to behavior modification.

## Limitation of the study

Because of its cross-sectional design, this study is unable to show the temporal relationship between cause and effect. Furthermore, this study focuses solely on awareness rather than the practice and attitude that women need to combat breast cancer.

**Table 3. Bivariate and multivariable analysis of breast cancer awareness and associated factors among reproductive-age women in West Badewacho Woreda, Hadiya Zone, South Ethiopia, 2020 (n = 364).**

| Variables | Categories | Aware n (%) | Unaware n (%) | COR(95%CI) | P-value | AOR(95%CI | P-value |
|---|---|---|---|---|---|---|---|
| Residence | Rural | 169(67.1) | 83(32.9) | 1.472(0.931,2.328) | 0.098 | | |
| | Urban | 65(58) | 47(42) | 1 | | | |
| Work status | Farmer | 37(50.7) | 36(49.3) | 0.131(0.055, 0.311) | <0.001 | | |
| | Housewife | 31(56.4) | 24(43.6) | 0.164(0.066, 0.407) | <0.001 | | |
| | Student | 47(52.2) | 43(47.8) | 0.139(0.060, 0.323) | <0.001 | | |
| | Merchant | 39(70.9) | 16(29.1) | 0.310(0.121, 0.791) | 0.014 | | |
| | Daily laborer | 17(85.0) | 3(15.0) | 0.720(0.172, 3.010) | 0.652 | | |
| | Gov't employee | 63(88.7) | 8(11.3) | 1 | | | |
| The educational level of women | cannot read and write | 50(60.2) | 33(39.8) | 0.366(.176, .759) | 0.007 | | |
| | read and write | 56(65.1) | 30(34.9) | 0.451(.216, .938) | 0.033 | | |
| | Primary | 37(59.7) | 25(40.3) | 0.357(.165, .774) | 0.009 | | |
| | Secondary | 33(54.1) | 28(45.9) | 0.284(.132, .615) | 0.001 | | |
| | higher or tertiary | 58(80.6) | 14(19.4) | **1** | | | |
| The educational level of husband | cannot read and write | 29(64.4) | 16(35.6) | 0.333(0.130,0.850) | 0.021 | 0.944(0.279,3.196) | 0.927 |
| | read and write(non-formal) | 35(53) | 31(47) | 0.207(0.088,0.490) | <0.001 | 0.351(0.122,1.008) | 0.052 |
| | Primary | 13(41.9) | 18(58.1) | 0.133(0.048,0.363) | <0.001 | 0.262(0.076,0.900) | 0.033* |
| | Secondary | 15(53.6) | 13(46.4) | 0.212(0.076,0.592) | 0.003 | 0.356(0.106,1.167) | 0.088* |
| | higher or tertiary | 49(84.5) | 9(15.5) | **1** | | 1 | |
| Income | 1311 | 140(57.9) | 102(42.1) | 0.409(0.250,0.669) | <0.001 | | |
| | >1311 | 94(77) | 28(23) | **1** | | | |
| Family history of BC | No | 151(58.1) | 109(41.9) | 0.351(0.205,0.601) | <0.001* | 0.281(0.132,0.594)* | 0.001* |
| | Yes | 83(79.8) | 21(20.2) | **1** | | 1 | |
| Having TV/Radio | No | 91(52.9) | 81(47.1) | 0.385(0.248,0.599) | <0.001* | 0.489(0.246,0.972)* | 0.041* |
| | Yes | 143(74.5) | 49(25.5) | **1** | | | |

Note:

*significantly associated, '1' reference group

## Conclusion

This study emphasized the importance of raising breast cancer awareness among women in the study area. The husband's educational status, family history of breast cancer, and having TV/Radio were significantly associated with breast cancer awareness. It is preferable to improve health-care programs that can reach all women regardless of their geographical location. Also, it is preferable to take advantage of opportunities to raise breast cancer awareness through health programs such as newspaper, local written and oral, radio, and television coverage of breast cancer signs and symptoms, risk factors, early diagnosis, prevention, and management.

It is also preferable to use cancer literacy programs at the national and state levels, as well as collaborations with community-level organizations and all levels of health-care delivery systems.

## Supporting information

**S1 Data.**
(DOCX)

## Acknowledgments

Our heartfelt thanks go to Hosanna College of Health Sciences' research and publication directorate, especially the research review board, for facilitating and fund this research work. We would also like to thank the Woreda Health Office, study participants, supervisor, and data collectors for their assistance during data collection. Finally, we would like to thank our advisor for his constant comments, advice, and suggestions.

## Author Contributions

**Conceptualization:** Mengistu Lodebo Funga.

**Data curation:** Mengistu Lodebo Funga.

**Formal analysis:** Mengistu Lodebo Funga.

**Funding acquisition:** Mengistu Lodebo Funga, Zerihun Damissie Dilebo, Anebo Getachewu Shuramo, Tessema Bereku.

**Investigation:** Mengistu Lodebo Funga, Anebo Getachewu Shuramo.

**Methodology:** Mengistu Lodebo Funga.

**Project administration:** Mengistu Lodebo Funga, Zerihun Damissie Dilebo, Anebo Getachewu Shuramo, Tessema Bereku.

**Resources:** Mengistu Lodebo Funga.

**Software:** Mengistu Lodebo Funga.

**Supervision:** Mengistu Lodebo Funga, Zerihun Damissie Dilebo, Anebo Getachewu Shuramo, Tessema Bereku.

**Validation:** Mengistu Lodebo Funga, Tessema Bereku.

**Visualization:** Mengistu Lodebo Funga, Zerihun Damissie Dilebo, Anebo Getachewu Shuramo, Tessema Bereku.

**Writing – original draft:** Mengistu Lodebo Funga, Zerihun Damissie Dilebo, Anebo Getachewu Shuramo, Tessema Bereku.

**Writing – review & editing:** Mengistu Lodebo Funga, Zerihun Damissie Dilebo, Anebo Getachewu Shuramo, Tessema Bereku.

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
