## [Decision Letter · Decision Letter 0]

24 Sep 2021

PONE-D-21-11416

ASSESSING BREAST CANCER AWARENESS ON REPRODUCTIVE AGE WOMEN IN WEST BADEWACHO WOREDA, HADIYYA ZONE, SOUTH ETHIOPIA: COMMUNITY BASED CROSS- SECTIONAL STUDY.

PLOS ONE

Dear Dr. Lodebo,

Thank you for submitting your manuscript to PLOS ONE. After careful consideration, we feel that it has merit but does not fully meet PLOS ONE’s publication criteria as it currently stands. Therefore, we invite you to submit a revised version of the manuscript that addresses the points raised during the review process.

We look forward to receiving your revised manuscript.

Kind regards,

Diwakar Mohan, MD

Academic Editor

PLOS ONE

Additional Editor Comments (if provided):

Dear Authors,

The major critique about your submission has been the quality of writing. This needs to be addressed using the services of a native English speaker. This rewrite will help the reviewers provide more focused feedback without getting lost / misunderstanding the text.

A minor issue regarding the statistical analysis is the lack of clarity on any adjustment for clustering of responses due to survey design. If you have adjusted for the design effect, please elaborate on how you have done so. if not, please redo all hypothesis testing with adjustment for clustering.

Journal Requirements:

A clean copy of the edited manuscript (uploaded as the new *manuscript* file).

3. In your Methods section, please provide additional information about the participant recruitment method and the demographic details of your participants. Please ensure you have provided sufficient details to replicate the analyses such as: a) the recruitment date range (month and year), b) a description of any inclusion/exclusion criteria that were applied to participant recruitment.

4. Please ensure you have discussed any potential limitations of your study in the Discussion, including study design, sample size and/or potential confounders.

5. In the Methods, please state:

- Why written consent could not be obtained

- Whether the Institutional Review Board (IRB) approved use of oral consent

- How oral consent was documented

For more information, please see our guidelines for human subjects research: https://journals.plos.org/plosone/s/submission-guidelines#loc-human-subjects-research

6. We note that the grant information you provided in the ‘Funding Information’ and ‘Financial Disclosure’ sections do not match. 

7. Thank you for stating the following financial disclosure: "no"

8. Thank you for stating the following in your Competing Interests section: "no"

9. We note that you have stated that you will provide repository information for your data at acceptance. Should your manuscript be accepted for publication, we will hold it until you provide the relevant accession numbers or DOIs necessary to access your data. If you wish to make changes to your Data Availability statement, please describe these changes in your cover letter and we will update your Data Availability statement to reflect the information you provide.

10 .Please upload a copy of Figure 1, to which you refer in your text on page 18. If the figure is no longer to be included as part of the submission please remove all reference to it within the text.

11. Please ensure that you refer to Figure 3 in your text as, if accepted, production will need this reference to link the reader to the figure.

12. Please include captions for your Supporting Information files at the end of your manuscript, and update any in-text citations to match accordingly. Please see our Supporting Information guidelines for more information: http://journals.plos.org/plosone/s/supporting-information. 

13. We noticed you have some minor occurrence of overlapping text with the following previous publication(s), which needs to be addressed:

- https://jebmh.com/assets/data_pdf/Phani_Madhavi_-_FINAL.pdf

- https://preprints.jmir.org/preprint/26691/submitted

- http://journals.sjp.ac.lk/index.php/ijms/article/view/3286

- http://pkjournal.org/?page_id=1520

The text that needs to be addressed involves the Introduction and the Results sections.

In your revision ensure you cite all your sources (including your own works), and quote or rephrase any duplicated text outside the methods section. Further consideration is dependent on these concerns being addressed.

Reviewers' comments:

Reviewer's Responses to Questions

**Comments to the Author**

1. Is the manuscript technically sound, and do the data support the conclusions?

Reviewer #1: Partly

Reviewer #2: Yes

2. Has the statistical analysis been performed appropriately and rigorously? 

Reviewer #1: No

Reviewer #2: I Don't Know

3. Have the authors made all data underlying the findings in their manuscript fully available?

Reviewer #1: Yes

Reviewer #2: Yes

4. Is the manuscript presented in an intelligible fashion and written in standard English?

Reviewer #1: No

Reviewer #2: No

5. Review Comments to the Author

Reviewer #1: GENERAL COMMENTS

• The author states that since breast cancer awareness is not well-documented among Ethiopian women, this study assesses awareness of the breast cancer and associated factors among reproductive-aged women in Hadiya Zone, West Badewacho Woreda of Ethiopia.

• The study shows that factors like husband’s education status, family history of breast cancer and having TV/Radio were significantly associated with awareness of breast cancer among reproductive-aged women.

• There is not a strong case made for the need for the study. Also, the authors are making a broad assumption of the Ethiopian situation without any references.

• Some of the methodological parts have not been clearly described, and there are some inconsistencies, as suggested in the “Materials and Methods” section.

• Some elaborations and clarifications can be made in the Discussion section , as suggested in the “Discussion” section.

• Some of the Recommended Items from the Strobe Checklist have not been illustrated.

o Variables in the Methods Section

o Limitations and Interpretations in the Discussion Section

• There are multiple problems with language, grammar and sentence structure, sometimes making the sentences difficult to comprehend. I have indicated many of them in the PDF as well, but there might be additional issues. It is suggested to re-write the manuscript considering language and sentence structure issues.

The Section Wise Details have been uploaded in a separate Word File.

Reviewer #2: The paper discusses an important aspect of public health - i.e public awareness about a serious but preventable disease that has a significant social burden. The methodology is good. I would like to recommend acceptance of the paper after it is completely re-written by a native english writer.

6. PLOS authors have the option to publish the peer review history of their article (what does this mean?). If published, this will include your full peer review and any attached files.

Reviewer #1: **Yes: **Seema Subedi

Reviewer #2: **Yes: **Dr Manikandan K

---

## [Author Response · Author response to Decision Letter 0]

10 Dec 2021

Response to editor and reviewers

Dear Editor and revewers

Thank you very much for your consideration of our manuscript and constructive, valuable and educational comments. We have considered each of them, and provide the changes affected below in turn. Where changes have not been made, we provide reasons for this to substantiate our view.

Thanks, we were rewriting our manuscript as you told.

Thanks, we also elaborated study design and design effect on the manuscript.

We used design effect for we have woreda, then kebeles, lastly gotti(mahandari). So, we have at least two stages. So, we used design effect.

Thank you

 We have updated the style, layout and naming of manuscript in line with the requirements of PLOS ONE. 

2. In your Methods section, please provide additional information about the participant recruitment method and the demographic details of your participants. Please ensure you have provided sufficient details to replicate the analyses such as: a) the recruitment date range (month and year), b) a description of any inclusion/exclusion criteria that were applied to participant recruitment.

Thank you

 All your requirements were incorporated in the manuscript like participants recruitments methods, recruitment date & range, inclusion and exclusion criteria.

3. Please ensure you have discussed any potential limitations of your study in the Discussion, including study design, sample size and/or potential confounders.

Limitation parts was added and considered.

4. Thanks a lot.

In the Methods, please state:

- Why written consent could not be obtained

- Whether the Institutional Review Board (IRB) approved use of oral consent

- How oral consent was documented

 This entire question was addressed in manuscript part in detail. Our subjects were human. So, ethical part was applied as research ethics guideline

 Thanks, we have considered these parts. Our funders were the institution but have no role in study design, data collection and analysis, decision to publish, or preparation of the manuscript

 Thank u, we also consider as your recommendation. So, the authors have declared that no competing interests exist.

7. We note that you have stated that you will provide repository information for your data at acceptance. Should your manuscript be accepted for publication, we will hold it until you provide the relevant accession numbers or DOIs necessary to access your data. If you wish to make changes to your Data Availability statement, please describe these changes in your cover letter and we will update your Data Availability statement to reflect the information you provide.

 Thank u. we will provide full data information for you, if you want any time.

8. Please upload a copy of Figure 1, to which you refer in your text on page 18

 Thank u. this was corrected in manuscript. It was figure 2. 

9. Please ensure that you refer to Figure 3 in your text as, if accepted, production will need this reference to link the reader to the figure.

 Also this part was corrected as figure 2. It was corrected in manuscript part

10. Please include captions for your Supporting Information files at the end of your manuscript, and update any in-text citations to match accordingly. Please see our Supporting Information guidelines for more information: http://journals.plos.org/plosone/s/supporting-information. 

Thanks a lot. We were considering it.

11. We noticed you have some minor occurrence of overlapping text with the following previous publication(s), which needs to be addressed:

 All this part was considered in manuscript. A manuscript was rewrite and cites all references.

Reviewer 1

Really we have great appreciation to your constructive and valuable comments and suggestion. In general, we have modified our manuscript completely as your comments and suggestion. For seek response we have some reaction on your comments

1. The author states that since breast cancer awareness is not well-documented among Ethiopian women, this study assesses awareness of the breast cancer and associated factors among reproductive-aged women in Hadiya Zone, West Badewacho Woreda of Ethiopia.

e.g. According to WHO report 2020

Cancer is a leading cause of death worldwide, accounting for nearly 10 million deaths in 2020 (1). The most common in 2020 (in terms of new cases of cancer) were:

 breast (2.26 million cases); 

 lung (2.21 million cases); 

 colon and rectum (1.93 million cases); 

 prostate (1.41 million cases); 

 skin (non-melanoma) (1.20 million cases); and

 stomach (1.09 million cases).

Between 30 and 50% of cancers can currently be prevented by avoiding risk factors and implementing existing evidence-based prevention strategies. The cancer burden can also be reduced through early detection of cancer and appropriate treatment and care of patients who develop cancer. Many cancers have a high chance of cure if diagnosed early and treated appropriately. 

 This means that a little known about breast cancer especially in rural women. Because of this they seek care in late stage of cancer and even some of them didn’t understand it was disease or not. Also, in our study area there was no study conducted. So, developing mother’s awareness to this disease may develop women to treat early and manage timely.

2. Some of the methodological parts have not been clearly described, and there are some inconsistencies, as suggested in the “Materials and Methods” section below.

 Thank u. this part was considered. We made it clear and consistent in manuscript as your comments.

3. Some elaborations and clarifications can be made in the Discussion section, as suggested in the “Discussion” section below.

 This has been revised according to your suggestion.

4. 

Abstract part

• Line 29 – The author has used both bivariable and multivariable logistic regression to determine the independent predictors. Please use “Binary and multivariable logistic regression analyses.”

Thank u. It is considered in manuscript part as bivariable and multivariable analysis. For general knowledge binary logistic regression is used for categorical varibles especially dependent variable is dichotomies. 

• Line 32 ( Result Section)- This is not a causal model of analysis. So, it would be better to say “Adjusting for other variables” instead of using the word confounders. This part is also considered in manuscript as you suggested.

• Line 37 ( Conclusion)- Please re-write the sentence “it needs to deal with awareness”. Here “it” is a vague word. Please clarify what the author wants to mention.

This was also considered in manuscript and re-written.

Introduction

• Line 57- Please re-write the sentence “Breast cancer is not only a disease of the developed countries although low and middle-income countries are also affected.”

This part is considered in manuscript as you suggest.

Materials and Methods

• Line 85- Please justify why the author used the 1.5 design effect. 

Thank u my reviewer. we used at least two stages lower in the sampling process to arrive at the final sampling unit. As a result, we used a design effect of 1.5 to multiply our sample size in order to minimize variability and detect the effect observed regarding breast cancer awareness. 

• Line 141- Please use the word “ bivariable” instead of bivariate. Also it would be better to use words uniformly; either “binary logistic regression”, or “bivariable logistic regression” in all the sections of the paper. 

It was considered as you suggested. Generally, bivariable for one dependent variable with one independent variable, for multivariable dependent variable with two or more independent variables.

• Line 200- Please use the word “multivariable analysis/regression” throughout the manuscript instead of the word “multivariate analysis/regression”. Please see the references for your reading . From one of the papers below-“Statistically speaking, multivariate analysis refers to statistical models that have 2 or more dependent or outcome variables, and multivariable analysis refers to statistical models in which there are multiple independent or response variables.” 

https://www.karger.com/article/FullText/345491#:~:text=The%20terms%20'multivariate%20analysis'%20and,outcome%20each%20time%20%5B1%5D. 

https://www.ncbi.nlm.nih.gov/pmc/articles/PMC3518362/

-This part is also corrected as you suggest

• Line 142- Please clarify why the author used the p-value of 0.25 instead of 0.05 in the bivariable model.

As you know that bivariable analysis is one outcome variable with one explanatory variable analysis. So, we used p-value <0.25 to get important variables that predict outcome variable due to bivariate analysis only compute with one variable. So, increase candidate variables for multivariable analysis. This mostly determine true predictor for outcome variable. As convection use criteria to select variables in bivariable analysis.

• Also, please be consistent with the use of the p-value of 0.25 or 0.05, as the p-value of 0.05 has been indicated in line 366 of Table 3 ( Bivariable Analysis).

Thank u. we used consistent throughout manuscript as your suggestion. But we used p-value <0.25 for bivariable analysis and p-value <0.05 for multivariable analysis. 

• Line 142-144- Please elaborate on why the author used the method “Back ward likelihood ratio of logistic regression”.

 Thank u. why we used this backward likelihood ratio of logistic regression from others types of stepwise due it has better than other like Starting with the full model has the advantage of considering the effects of all variables simultaneously. Also, important in case of collinearity. Due to force to keep them all in the model.

• Line 145- “AOR” stands for Adjusted Odds Ratio, not just Odds Ratio.

It was considered in manuscript part

• Line 147-148- Please clearly indicate if this statement “P-value < 0.05 was considered as statistically significant in the study” is for multivariable model.

Yes. It was considered in manuscript as you suggest

• According the to the STROBE checklist, all the VARIABLES should be clearly defined in the Method Section. 

o The outcome variable is the division of respondent’s score by the mean score. This sounds a bit arbitrary – what would happen if the mean is skewed because of outlier values? The usual method is to use the median – using quantile regression as the multivariable method. Please clarify on this.

Thank u. we used distribution of the outcome variable before mean score calculation. So, our data was normally distributed. 

o The knowledge score is not well defined – what were the questions that went into the score? How much was the scale and what were the minimum and maximum? All these need to be part of the methods description of the variables.

It was considered in manuscript part specifically in operational definition part

o Please Insert a VARIABLE Section and define all the variables clearly. 

It was inserted in manuscript part as you suggested.

Result

• Please check the tense in this section as indicated in General Comments section.

It was checked and re arranged in document.

• Line 196- Please be consistent with the use of the p-value of 0.25 or 0.05, as it is indicated p-value of 0.05 in line 366 of Table 3 ( Bivariable Analysis).

Thanks a lot. We used and try to clarify in above section we used p-value <0.25 in bivariate analysis to get candidate variables and p-value <0.05 for multivariable analysis. But, we use convection and rule of tump specially in bivariable analysis.

• Line 201-203: It would be better to include these sentences in the Data Analysis Section.

It was considered in document

• In the Result Section, as well as in the Tables, two significant digits after the decimal point are enough.

This part was also considered in document as you recommended

Discussion

• Line 222-224- “This difference might be due to sample size, respondent’s inclusion criteria, and data collection methods, total number of items they were used to ask the respondents, construction and compute of the awareness item”. The author could specify one or two instances of how the different methodological aspects like sample size or total number of items used to ask respondents could have led to lower prevalence of breast cancer awareness in Eastern China (18.6%) and Saudi Arabia (39.7%) , as compared to this study (64.3%).

Prevalence -The number of cases of a disease (or people with a particular characteristic) existing in a specified population at a given point in time. So, if sample size decrease cases probability to get cases might be decrease. Also, item they used to ask respondents different from us or categorization way may influence prevalence due to understandability of item to respondents affect responses to item. This mean that item number increase the clarity to answer increase.

• According to Strobe Checklist, there needs to be a section with LIMITATIONS and INTERPRETATION in the discussion. Please add that.

Considered and incorporated in manuscript part especially limitation. But interpretation was incorporated in conclusion part.

Conclusion

• This section requires re-writing as indicated in the General Comments Section.

This section was re-written in manuscript as you suggested.

COMMENTS ON THE TABLE

Table 1

• Check the case ( higher/lower) of the letters of the variables

It was checked and corrected in manuscript.

Table 2

• The author put asterisk sign in the footnote as “*more than one option is reported by a participant”, but the asterisk sign is not indicated anywhere in the table.

This part was corrected and asterisk was putted in correct place on table as you mentioned.

Table 3 and Table 4

• The author could show the information in Table 3 and Table 4 ( Crude Odds Ratio and Adjusted Odds Ratio) in a single table for better presentation and clarity.

Considered and merged in one table as your suggestion 

• Line 366- “ Note: * p value of less than 0.05 in bivariate analysis”. The author had described in the Method Section (Line 142) that in the bivariable analysis, they used the p-value of 0.25. In this Line 366, they indicate p value of 0.05. Please apply consistent methods, and clarify.

It was mentioned above why we used p-value less than 0.25 and p- value less than 0.05.

• Line 366- Also check the appropriate use and label of asterisk sign “ * ” defined as “p value of less than 0.05 in bivariate analysis” , since this asterisk sign has also been used in “<0.001*”. 

Why we used <0.001*, for those p-value was 0.000 and to show strong relationship between dependent and independent variables.

Reviewer 2

Really we have great appreciation to your constructive and valuable comments and suggestion. In general, we have modified our manuscript completely as your comments and suggestion

---

## [Decision Letter · Decision Letter 1]

16 May 2022

PONE-D-21-11416R1Assessing breast cancer awareness on reproductive age women in West Badewacho Woreda, Hadiyya Zone, South EthiopiaPLOS ONE

Dear Dr. Lodebo,

Thank you for submitting your manuscript to PLOS ONE. After careful consideration, we feel that it has merit but does not fully meet PLOS ONE’s publication criteria as it currently stands. Therefore, we invite you to submit a revised version of the manuscript that addresses the points raised during the review process.

We look forward to receiving your revised manuscript.

Kind regards,

Bijaya Kumar Padhi, PhD, MPH

Academic Editor

PLOS ONE

Journal Requirements:

Reviewers' comments:

Reviewer's Responses to Questions

**Comments to the Author**

1. If the authors have adequately addressed your comments raised in a previous round of review and you feel that this manuscript is now acceptable for publication, you may indicate that here to bypass the “Comments to the Author” section, enter your conflict of interest statement in the “Confidential to Editor” section, and submit your "Accept" recommendation.

Reviewer #3: All comments have been addressed

2. Is the manuscript technically sound, and do the data support the conclusions?

Reviewer #3: Yes

3. Has the statistical analysis been performed appropriately and rigorously? 

Reviewer #3: Yes

4. Have the authors made all data underlying the findings in their manuscript fully available?

Reviewer #3: Yes

5. Is the manuscript presented in an intelligible fashion and written in standard English?

Reviewer #3: Yes

6. Review Comments to the Author

Reviewer #3: The study “ASSESSING BREAST CANCER AWARENESS ON REPRODUCTIVE AGE WOMEN IN WEST BADEWACHO WOREDA, HADIYYA ZONE, SOUTH ETHIOPIA: COMMUNITY BASED CROSS- SECTIONAL STUDY” is significant in health promotion in NCD. However, the author need to address following minor suggestion.

Results

Table 1 Remove total (n, %) from each variable.

Remove FIGURE 2 AWARENESS OF BREAST CANCER AMONG REPRODUCTIVE-AGE WOMEN IN WEST BADEWACHO WOREDA, HADIYA ZONE, SOUTH ETHIOPIA,2020 (N=364). Describe it in text.

FIGURE 3 SOURCE OF INFORMATION FOR THE AWARENESS, present a simple bar diagram.

Discussion

Revise the discussion section, focused more on what is the existing health promotion program in the country. How this study findings will be useful to promote breast cancer screening in the country.

7. PLOS authors have the option to publish the peer review history of their article (what does this mean?). If published, this will include your full peer review and any attached files.

Reviewer #3: **Yes: **Dr. Krushna Chandra Sahoo

---

## [Author Response · Author response to Decision Letter 1]

30 May 2022

Response to reviewer 

Dear Peer Reviewer,

Thank you very much still for your consideration of our manuscript and constructive, valuable and educational comments. We have considered each of them, and provide the changes affected below in turn. Where changes have not been made, we provide reasons for this to substantiate our view.

Really, we have great appreciation to your constructive and valuable comments and suggestion. In general, we have modified our manuscript completely as your comments and suggestion. For seek response we have some reaction on your comments

Result part

1. Remove FIGURE 2 AWARENESS OF BREAST CANCER AMONG REPRODUCTIVE-AGE WOMEN IN WEST BADEWACHO WOREDA, HADIYA ZONE, SOUTH ETHIOPIA,2020 (N=364). Describe it in text.

 Thank u. this part was considered as you commented.

2. FIGURE 3 SOURCE OF INFORMATION FOR THE AWARENESS, present a simple bar diagram.

 This has been revised according to your suggestion.

Discussion

Revise the discussion section, focused more on what is the existing health promotion program in the country. How this study findings will be useful to promote breast cancer screening in the country.

Thank u. we were discussed most of our findings with other findings conducted in other part of countries. As result we focused on contemporary issues. As you know that if women aware for breast cancer, they will be screened for breast cancer regularly. if not, the chance of screening is low. our aim was to identify level of awareness of reproductive aged women rather than screening awareness. Some how, it was raised on significant of study part and conclusion.

---

## [Editor Report · Decision Letter 2]

8 Jun 2022

Assessing breast cancer awareness on reproductive age women in West Badewacho Woreda, Hadiyya Zone, South Ethiopia; Community based cross- sectional study.

PONE-D-21-11416R2

Dear Dr. Lodebo,

We’re pleased to inform you that your manuscript has been judged scientifically suitable for publication and will be formally accepted for publication once it meets all outstanding technical requirements.

Kind regards,

Bijaya Kumar Padhi, PhD, MPH

Academic Editor

PLOS ONE
---

## [Editor Report · Acceptance letter]

13 Jun 2022

PONE-D-21-11416R2 

Assessing breast cancer awareness on reproductive age women in West Badewacho Woreda, Hadiyya Zone, South Ethiopia; Community based cross- sectional study. 

Dear Dr. Funga:

I'm pleased to inform you that your manuscript has been deemed suitable for publication in PLOS ONE. Congratulations! Your manuscript is now with our production department. 

Kind regards, 

on behalf of

Dr. Bijaya Kumar Padhi 

Academic Editor

PLOS ONE